# Whole Goat Milk as a Source of Fat and Milk Fat Globule Membrane in Infant Formula

**DOI:** 10.3390/nu12113486

**Published:** 2020-11-13

**Authors:** Sophie Gallier, Louise Tolenaars, Colin Prosser

**Affiliations:** Dairy Goat Co-operative (N.Z.) Limited, Hamilton 3240, New Zealand; Louise.tolenaars@dgc.co.nz (L.T.); Colin.prosser@dgc.co.nz (C.P.)

**Keywords:** milk fat, milk fat globule membrane, goat milk, infant formula, infant nutrition

## Abstract

Cow milk is the most common dairy milk and has been extensively researched for its functional, technological and nutritional properties for a wide range of products. One such product category is infant formula, which is the most suitable alternative to feed infants, when breastfeeding is not possible. Most infant formulas are based on cow milk protein ingredients. For several reasons, consumers now seek alternatives such as goat milk, which has increasingly been used to manufacture infant, follow-on and young child formulas over the last 30 years. While similar in many aspects, compositional and functional differences exist between cow and goat milk. This offers the opportunity to explore different formulations or manufacturing options for formulas based on goat milk. The use of whole goat milk as the only source of proteins in formulas allows levels of milk fat, short and medium chain fatty acids, *sn*-2 palmitic acid, and milk fat globule membrane (MFGM) to be maximised. These features improve the composition and microstructure of whole goat milk-based infant formula, providing similarities to the complex human milk fat globules, and have been shown to benefit digestion, and cognitive and immune development. Recent research indicates a role for milk fat and MFGM on digestive health, the gut–brain axis and the gut–skin axis. This review highlights the lipid composition of whole goat milk-based infant formula and its potential for infant nutrition to support healthy digestion, brain development and immunity. Further work is warranted on the role of these components in allergy development and the advantages of goat milk fat and MFGM for infant nutrition and health.

## 1. Breast Milk and Breast Milk Substitutes

Human breastmilk is the optimal source of nutrition for the human infant. While it is recommended to exclusively breastfeed to 6 months of age [1], infant formulas are the most suitable breast milk substitutes when breastfeeding is not possible. Cow milk ingredients have traditionally been used as the source of proteins in the manufacture of infant, follow-on and young child formulas. The main source of lipids in formulas is commonly a blend of vegetable oils, and lactose, vitamins and minerals are added to match the macro- and micronutrient composition of human milk as outlined in local regulations [2,3]. While whole cow milk can be used in the formulation of infant formulas, it is more common to use skim milk and whey protein powdered ingredients to reach a whey:casein ratio of 60:40, as in mature human milk, while providing an adequate amino acid supply within the allowed protein levels in the regulations [4]. However, the whey protein adjustment limits the amount of milk fat in the final infant formula. Other sources of cow milk fat could be added such as liquid or powdered cream and anhydrous milk fat, but these ingredients are usually costly or not available in paediatric grade [3].

Human milk lipids are present as milk fat globules with a complex structure of triglyceride droplets stabilized by a trilayered membrane, the milk fat globule membrane (MFGM) [5]. Human milk fat has a unique fatty acid profile, with a high level of palmitic acid [6]. To match the palmitic acid concentration of human milk, infant formula manufacturers need to add unmodified or structured palm oil or milk fat [2,3]. In the last decade, there has been increased criticism over the use of palm-derived products and a push from consumers to limit the use of such ingredients [7]. In addition, vegetable oils, and in particular palm oil, contain some contaminants such as 2- and 3-monochloropropane-1,2-diols and their fatty acid esters formed during the oil refining process [8]. This has led to a resurgence in the use of milk fat for infant formula products, particularly in Europe [3].

## 2. Infant Formula Made from Goat Milk

While the use of animal milk to feed infants has a long tradition, infant formula products are usually based on cow milk ingredients or plant-based ingredients, e.g., soy or rice protein isolate as a source of proteins and maltodextrin or other sugars as a source of carbohydrates [2]. Due to increased consumer demand for alternatives to cow milk [9] and the recognition of the presence of anti-nutritional factors, such as trypsin inhibitor and phytate, phytoestrogens and undesirable taste in plant-based products [10,11], goat milk has been reintroduced in pediatric nutritional products in the last 30 years. Goat milk is perceived to be easier to digest than cow milk, making it beneficial for infant nutrition [12,13,14]. Milk from goats with a low α_s1_-casein genotype is favoured for manufacture of infant formulas, as α_s1_-casein is a predominant milk allergen [15] and the milk forms a softer curd in acidic environment such as in the stomach [16,17]. Other factors such as nutritive value (absence of A1 β-casein in goat milk [18]) and an association with tradition [19,20] make goat milk more attractive to some consumers.

As with traditional infant formulas based on cow milk protein ingredients, some formulas based on goat milk are whey-adjusted, i.e., using goat skim milk and whey protein ingredients to reach a whey:casein ratio of 60:40 and therefore using mainly vegetable oils as the source of lipids [14]. This review will focus on the infant, follow-on and young child formulas based on whole goat milk without adjustment of the whey:casein ratio (i.e., maintaining the natural 20:80 whey:casein ratio of whole goat milk) and with clinical evidence that it supports healthy growth and development of the infant [21,22,23]. The use of whole goat milk has two main advantages, it allows to match more closely the diverse composition of human milk fat but also its complex structure.

## 3. Whole Goat Milk Fat to Supply a Variety of Fatty Acids and *sn*-2 Palmitate

The α_s1_-casein genotype has an impact on the fatty acid profile of goat milk, with the low α_s1_-casein genotypes associated with increased level of unsaturated fatty acids [24]. Goat milk and cow milk have a relatively comparable total fat content and fatty acid profile [25,26]. SCFAs, in particular C4:0, are not always detectable or reported for human milk (Table 1), however a few studies have reported levels of C4:0 in lower concentrations than in cow and goat milk fat ([27,28]; Table 1). Fat from goat milk contains a higher amount (15%–18%) of the MCFAs C6:0, C8:0 and C10:0 compared to cow milk (5%–9%) [26]. As MCFAs are more readily released and absorbed in the gastrointestinal tract (GIT), this unique composition may contribute to a greater digestibility of goat milk fat compared to cow milk fat [26]. Cow milk contains on average slightly more palmitic acid than goat milk (Table 1).

Human milk has more oleic, docosahexaenoic (DHA), arachidonic acid (ARA), linoleic and α-linoleic acids than goat or cow milk (Table 1). Hence, it is not possible to use goat or cow milk fat as the only source of lipids for infant formulas. Therefore, vegetable oils and oils rich in DHA and ARA must be added to match the fatty acid profile of human milk. The fatty acid profile of two examples of infant formulas made from whole goat milk, one with a 48% and one with 55% of lipids as milk fat, is given in Table 1. By using a combination of vegetable oils and milk fat from whole goat milk, levels of unsaturated fatty acids comparable to human milk can be achieved. Most importantly the use of milk fat in infant formula provides a more complex fatty acid profile similar to that found in human milk (Table 1), with increased levels of short and medium chain fatty acids (SCFAs and MCFAs, respectively), presence of fatty acids typical of milk fat such as C15:0, C16:1 *n-7*, C17:0, conjugated linoleic acid (CLA) and branched-chain fatty acids (BCFAs), and comparable levels of palmitic acid. A similar compositional effect can be obtained by using milk fat from cow milk-derived ingredients to manufacture cow milk-based infant formulas (Table 1). Indeed, more common cow milk-based infant formulas are manufactured using skim milk and whey protein powders blended with vegetable oils with or without palm oil. As the lipid composition of infant formulas is influenced by the source of vegetable oils and the type of dairy ingredients used in the formulation, the fatty acid profile of three cow milk-based infant formulas is given for comparison in Table 1.

Early reports indicate that goat milk contains 280 mg β-keto acids/kg of fat, in particular those of 16 and 18 carbon chain length [30]. In addition, rapid absorption and β-oxidation of dietary MCFAs result in the release of plasma ketones, a source of energy and acetyl-CoA for the brain of the growing infants [31]. Breastfed infants have higher levels of plasma ketones than formula-fed infants [31]. Thus, an infant formula made from whole goat milk provides a source of MCFAs and β-keto acids, which may contribute to brain development. In addition, goat milk contains BCFAs, such as 4-methyl- and 4-ethyl-octanoic acid [32]. BCFAs are also found in human milk and their concentration is influenced by maternal diet [33]. BCFAs are essential to bacterial membrane functions and have been shown in limited animal and human studies to alter the gut microbiota [33]. CLA (*c*9, *t*11) levels in goat milk and infant formula based on whole goat milk are within the same range as found in human milk (Table 1). A study looking at CLA levels in human milk and a range of cow milk-based formulas found that human milk contained significantly more CLA and some infant formulas did not contain any detectable levels [34]. While CLA is considered an anticarcinogen, its role in infant nutrition remains unclear.

As seen in Table 1, both goat milk and cow milk are good sources of palmitic acid. The use of vegetable oils as the only source of lipids in infant formulas results in levels of palmitic acid below (7.7%), when palm oil is not used, or similar (26.3%), when palm oil is used, to that of human milk (15.4%–29.0%) (Table 1). Depending on the level of milk fat used, the use of whole goat milk as a source of milk fat in infant formula results in infant formulas with a level (12.3%–17.0%) of palmitic acid close to the lower end of the concentration in human milk (15.4%–29.0%) (Table 1). Similarly the use of cow milk fat in infant formulas results in comparable levels of palmitic acid (Table 1).

Human milk is quite unique in the way that up to 70% of its palmitic acid is found in the *sn*-2 position of the triglyceride molecules [6]. Reports on the *sn*-2 palmitic acid content in human milk (Table 2) vary in the literature partly due to different analytical methods used but the *sn*-2 palmitic acid content does not seem affected by maternal diet or geographical location [31]. Palm oil is commonly added in infant formulas to reach a palmitic acid level within the range found in human milk (Table 1). However most of the palmitic acid in palm oil is in the *sn*-1,3 position, unless it has been enzymatically or chemically interesterified to result in structured palm oil rich in 1,3-dioleyl-2-palmitoylglycerol, also called OPO or Betapol [7]. Cow milk-based infant formulas have therefore a variable *sn*-2 palmitic acid content depending on the type of added vegetable oils (Table 2). Goat milk and cow milk are natural sources of *sn*-2 palmitic acid (Table 2). By using whole goat milk without modification of the whey:casein ratio, an infant formula can contain up to 31% of palmitic acid in the *sn*-2 position, so about half the level in human milk (Table 2), but more than the level in infant formulas (as low as 8%) manufactured with vegetable oils only [2]. The advantage of the stereospecifity of palmitic acid for infant nutrition, beyond mimicking the triglyceride structure of human milk fat, will be discussed later.

## 4. Whole Goat Milk as a Source of Cholesterol

Human, cow and goat milk fats contain similar levels of cholesterol (Table 3), present in both the core and the membrane of fat globules and about 90% in the free state (vs. 10% as cholesterol esters) [30,36]. Infant formulas, with vegetable oils as the only source of lipids, contain very little cholesterol deriving from residual levels in skim milk and whey protein ingredients [3]. Raising the level of cholesterol of infant formula close to that of human milk can be achieved by adding anhydrous milk fat or cream as sources of lipids but also by adding MFGM-rich ingredients [3,37,38,39]. In Timby et al. [37], a concentration of 80 mg of cholesterol/L was achieved by supplementing the experimental infant formula with an MFGM-rich whey ingredient. Claumarchirant et al. [38] analysed a range of commercial infant formulas and reported levels of total animal sterols between 17.1 and 54.6 mg/L and cholesterol between 14.6 and 51 mg/L, with the highest levels found in infant formulas supplemented with cow milk fat and an MFGM-rich whey ingredient. Using unmodified whole goat milk as a source of native milk fat globules (i.e., core and MFGM) is another way to increase the level of cholesterol in infant formula to about half the average cholesterol level of human milk (Table 3).

## 5. The Secretion of Goat Milk

Another advantage of using fresh unmodified whole goat milk is the addition of native milk fat globules with MFGM. The secretion mechanism of fat globules in cows, goats and humans have been shown to be similar, involving secretion of fat globules surrounded by the secretory cell membrane [30,42]. While the structure of MFGM in milks of most species is similar, a few reports showed an increased presence of cytoplasmic crescents attached to goat milk fat globules compared to cow milk fat globules, a thicker goat MFGM and a more filamentous human MFGM glycocalyx [30,43,44,45]. The amount of MFGM polar lipids appears higher in mid-lactation milk from goats with a null genotype for α_s1_-casein than those with a strong genotype for α_s1_-casein, potentially explained by a smaller fat globule diameter in the former [46]. An upregulation of MFGM lactadherin and stomatin was observed in early-lactation milk from goats with a null genotype for α_s1_-casein. Furthermore, Cebo et al. [46] reported an increase in total phospholipids, in particular phosphatidylinositol (PI) and sphingomyelin (SM), in milk fat from goats with a null genotype for α_s1_-casein.

While the mechanism of milk secretion has been studied for over 150 years and a clear consensus has not been reached, the secretion of goat milk and human milk is considered to be apocrine, while cow milk is secreted through a merocrine process [43]. The presence of large numbers of cytoplasmic fragments in goat milk and human milk has been hypothesised to be due to a greater emphasis on apocrine secretion [42,44]. Cellular fragments are also common in human milk but are less abundant in cow milk [47], suggesting goats and humans share similar secretion mechanisms. In goats, at least, there is evidence that the normal secretion process for casein proteins is perturbed at low α_s1_-casein levels, resulting in proteins being secreted by the apocrine process [16,48].

Apocrine secretion results in cytoplasmic material from the mammary cell being released in the milk in the alveolar lumen and an increased number of milk fat globules with cytoplasmic crescents [16,47]. Cytoplasmic crescents are known to enclose various vesicles and miRNA, which may benefit the infant’s immune system [5]. In several studies, it was demonstrated that only 1% of cow milk fat globules were associated with cytoplasmic crescents, while up to 5% of goat milk fat globules and 7.2% of human milk fat globules were associated with cytoplasmic crescents [43,44,45]. Despite identification of cytoplasmic material in the milk and associated with the MFGM decades ago, their role for the health of the offspring or as marker of the health of the mammary gland remains undetermined.

## 6. Whole Goat Milk as a Source of MFGM Polar Lipids

As with cow milk, it is estimated that about 60% of goat milk phospholipids are associated with the MFGM and 40% are found in skim milk, which migrate from the MFGM after secretion of the milk fat globules into the alveolar lumen of the mammary cell [30]. Goat milk contains 30–40 mg phospholipids/100 g milk or 8–10 mg phospholipids/g fat. Similar to human milk and cow milk (Table 4), the main goat milk phospholipids are phosphatidylcholine (PC), phosphatidylethanolamine (PE), SM, phosphatidylserine (PS) and PI. Phospholipid levels in human, goat and cow milk vary with genetics, lactation stage and diet, but reports in the literature also vary with methods of extraction and analysis [26,49]. Phospholipid measurement with ^31^P-NMR allows the distinction between SM and dihydrosphingomyelin (DHSM) unlike chromatographic methods but SM and DHSM are commonly summed [50]. By using whole goat milk to manufacture infant formula, it is possible to supply phospholipids, and in particular SM, to levels and proportion close to that of human milk (Table 4). Infant formulas based on vegetable oils as the main source of lipids or not enriched with MFGM usually will contain a limited amount of phospholipids, mainly derived from soy or sunflower lecithin used as an emulsifier [3,39], therefore with lower levels of SM and PS and higher proportion of PC, PE and PI than in human milk [49].

There are only limited reports on the other minor MFGM lipids in goat milk, such as cerebrosides (2.50 mg/100 mL milk), glucosyl ceramide (0.97 mg/100 mL) and lactosyl ceramide (1.53 mg/100 mL) [30]. Total goat milk gangliosides, sialic acid-rich glycolipids mainly present in the MFGM, have only been reported as lipid-bound sialic acid (LBSA) and individual gangliosides separated by thin layer chromatography [53,54]. Gangliosides were reported to be unaffected by heat treatment [53]. The level of total goat milk gangliosides is about 2.5 mg/kg colostrum (day 1) and 0.4 mg/kg mature milk, about 2–10 times lower than in cow milk depending on the stage of lactation. GM3, GD3 and GT3 represent 66%–92% of LBSA and contain sialyllactosylceramide [53]. The ratio of N-glycolylneuraminic acid (Neu5Gc) to N-acetylneuraminic acid (Neu5Ac) increased from 30% to 60% in the first 2 months of lactation [53]. Neu5Gc and Neu5Ac sialic acid contents in goat milk was shown to vary across lactation, with the total sialic content varying between 12.2 and 63.1 mg/100 g milk and a constant 60:40 Neu5Gc:Neu5Ac ratio [55]. Unfortunately, ganglioside levels measured using more accurate techniques or taking into consideration the increased proportion of Neu5Gc in goat milk are lacking. The mean total gangliosides (mainly measured as GD3: aNeu5Ac(2–8)aNeu5Ac(2–3)bDGalp(1–4)bDGlcp (1–1)Cer) in growing-up milk powders (also known as stage 3 or young child formulas) based on cow or goat milk varied from 0.03 to 11.0 mg/100 g powder [56]. Goat milk-based powders had the lowest ganglioside levels (0.03–0.2 mg/100g). However, Neu5Gc-based gangliosides were not measured. Indeed, analytical methods are usually tailored to quantify milk Neu5Ac-gangliosides, which is the only type present in human milk and the main type in cow milk. Neu5Gc and Neu5Ac differ by a single oxygen and are the two major mammalian sialic acid forms in glycoproteins and glycolipids. It is unclear what impact xeno-immunization resulting from consumption of dietary Neu5GC has on human health [57].

## 7. Whole Goat Milk as a Source of MFGM Proteins

The MFGM is also a rich source of membrane specific proteins, including glycosylated proteins and enzymes [58]. The human and bovine MFGM proteomes have been extensively studied but the goat MFGM proteome is being increasingly studied [59,60,61]. Quantitative analysis remains challenging and mainly focuses on relative abundance and function of proteins.

While the caprine MFGM proteome is less diverse than the bovine MFGM proteome (520 vs. 1012 proteins, respectively), they are both dominated by the same eight MFGM proteins: mucin 1 (MUC1), butyrophilin (BTN), xanthine oxidase, lactadherin (also called periodic acid Schiff 6/7 or MFGE8), fatty acid binding protein, perilipin-2, CD36 and mucin 15 [26,59]. O-glycans were found on goat MFGM fatty acid synthase and xanthine oxidase [59]. A comparison of the MFGM proteins between cow, goat, sheep, camel and horse species revealed some similarities but also some differences, in particular for butyrophilin and lactadherin, likely due to varying glycosylation [62]. Goat MUC1 is larger than MUC1 form sheep and cow milk. Goat BTN (67kDa) is slightly larger than cow milk BTN (64 kDa) but smaller than horse milk BTN (70 kDa). Goat lactadherin is made of one single polypeptide chain of 54 kDa unlike cow lactadherin with two polypeptide chains. Lactadherin, an antiviral protein involved in the maintenance of the intestinal epithelium, is also more abundantly expressed in goat milk than in human milk [63]. Goat lactotransferrin, an antipathogenic protein in the MFGM, is expressed at a lower level than its human counterpart but their structures are highly homologous [63]. Using whole goat milk to manufacture infant formula is a way to potentially improve the protein composition of infant formula by supplying MFGM proteins, known to play a protective role against pathogens, as described later. Unfortunately, qualitative and quantitative methods, including isolation or extraction and analysis, to measure MFGM proteins have not been applied to infant formulas yet.

## 8. The Relevance of the Fat Microstructure in Infant Nutrition

With the amount of research over the last two decades, we now have a better understanding of the importance of food beyond nutrient content and supply and the role of the food matrix as a key driver of food digestion, nutrient delivery, and digestive and physiological outcomes [36,64]. Human milk is more than just the sum of its nutrients; some nutrients are packaged in a way to enhance nutrient digestion, absorption or delivery within the infant GIT [36]. As such, the milk fat globule provides more than just energy, with the MFGM structural complexity likely optimized for the digestion of milk fat, the metabolism of MFGM components and their physiological effects [36].

Transmission electron microscopy was used to reveal the fat and protein structures in raw whole goat milk (Figure 1), whole goat milk powder (Figure 2) and infant formula based on whole goat milk with unmodified whey:casein ratio (Figure 3). As mentioned previously, goat MFGM in raw goat milk can have a thickness of up to 50-100 nm and be thicker (Figure 1A) than commonly observed for MFGM of other species [42] and may enclose cytoplasmic crescents (Figure 1B). Figure 1A,B also show that casein micelles in raw goat milk vary in size and are less than 500 nm in diameter.

Whole goat milk powder is obtained by pasteurization, homogenization and spray-drying of whole goat milk. As for cow milk, this results in interaction of milk proteins with the MFGM, disruption of the MFGM, and reduction of the size of the milk fat globules [64]. The size distribution of the casein micelles in whole goat milk powder (Figure 2) was heterogenous and similar to that in raw whole goat milk (Figure 1). MFGM fragments were present in the aqueous phase of goat milk, while some goat milk fat globules were mainly or partially stabilized by the MFGM (Figure 2). Processing also resulted in interfaces stabilized by milk proteins, and an increased number of milk fat globules with interfaces stabilized by caseins were observed, as compared with raw goat milk fat globules (Figure 1).

Manufacturing infant formula with whole goat milk and some vegetable oils involving pasteurization and homogenization leads to an oil droplet structure (Figure 3) similar to the structure of a standard (60:40 whey:casein ratio) infant formula oil droplet [5], despite having a 20:80 whey:casein ratio. Indeed, the oil droplets (Figure 3) are stabilized by a thin layer, likely composed of whey proteins and milk phospholipids from whole goat milk, and a few adsorbed small casein micelles. However, a few droplets (Figure 3B) presented a slightly thicker interface, composed of remnants of the native MFGM from milk fat globules in whole goat milk. In addition, as seen in Figure 3A,B, some intact MFGM fragments of various lengths were observed in the aqueous phase. Cytoplasmic crescents and their content were still found associated with MFGM fragments (Figure 3B). The presence of MFGM fragments, with a near-intact structure in the aqueous phase or at the interface of formula droplets and associated with cytoplasmic crescents, in infant formula based on whole goat milk may help delivering key MFGM components to the infant by matching more closely the composition but also the structure of human milk fat globules, as discussed later.

## 9. Role of Milk Fat and MFGM in Infant Nutrition, Growth and Development

To be suitable as the sole source of nutrition for infants up to the age of 6 months and in combination with complementary foods thereafter, formulas based on goat milk must adhere to similar strict compositional regulations as formulas based on cow milk to supply suitable levels of proteins, carbohydrates, fat, vitamins and minerals for optimal growth of the infant [30]. As the clinical research in pediatric populations consuming goat milk-based infant formulas is scarce at present, most of our knowledge on the role of milk fat to support healthy growth and development of the infant is based on studies with breastfed infants and infants fed cow milk-based formulas. The reader is referred to several recent reviews of preclinical and clinical research on the role of milk lipids on digestion, metabolic programming, immunity and cognitive development in early life [3,36,39,49,64,65]. The following section provides a snapshot of the potential for whole goat milk-based infant formula to support lipid digestion and metabolism and the development of the brain, cognition and immunity, and its possible influence on the development of allergy in early life.

### 9.1. Influence of Lipid Composition and Structure on Lipid Digestion and Metabolism

Human milk lipids are considered to be digested and absorbed more easily than infant formula lipids [66,67]. This is not only because of its specific composition, but also its triglyceride structure and microstructure of the milk fat globule [36].

#### 9.1.1. Kinetics of Digestion

In early life, gastric lipase plays a key role in the digestion of lipids due to low output of intestinal digestive enzymes. Gastric lipase initiates the digestion of triglycerides at the *sn*-1 or *sn*-3 position in the stomach and remains active under intestinal conditions [64]. SCFAs and MCFAs at the *sn*-1,3 positions are quickly released in the stomach, absorbed as free fatty acids (FFAs) in the intestine and rapidly degraded by mitochondrial β-oxidation in the liver [6]. Fatty acids at the *sn*-2 position are released in the intestine by combined action of lipases from pancreatic secretion and gastric lipase (and bile salt-stimulated lipase in human milk) as *sn*-2 monoglycerides (MAG), which are more polar than FFAs and thus more easily absorbed. LCFAs in the *sn*-1,3 positions are released as free fatty acids, solubilized into bile salt micelles and absorbed in the small intestine. 

Vegetable oils do not contain SCFAs and MCFAs (with the exception of coconut oil) nor odd-chain fatty acids, therefore having a less diverse fatty acid and triglyceride range than human, goat and cow milk ([3]; Table 1). In animal models, consumption of MCFAs in early life reduced the negative impact of a high-fat diet, in particular insulin sensitivity and fat accumulation, in adulthood [3]. In a prospective birth cohort, human milk SCFA concentration was associated with a beneficial effect on weight gain, adiposity and its related metabolic functions up to 2 years of age [27].

Using whole goat milk as a source of goat milk fat results in a more diverse fatty acid profile in infant formulas compared with formulas including only vegetable oils as the source of lipids. This diversity may alter lipid digestion kinetics in the stomach and small intestine to provide a sustained delivery of energy. In particular, the supply of goat milk SCFAs and MCFAs, easily digested and absorbed, may provide a rapid source of fuel as does human milk [6].

#### 9.1.2. Triglyceride Structure

Although saturated fatty acids are considered to have undesired health effects in adults, palmitic acid is a key component of membrane, secretory and transport lipids and plays an essential part in protein palmitoylation and signal molecules [31]. The replacement of milk fat by vegetable oils and avoidance of the use of palm oil leads to formulas low in palmitic acid, which may modify the composition of developing tissues and lowers plasma cholesterol levels [31].

While MCFAs, and in particular SCFAs, have a good solubility in water under gastric and intestinal conditions, saturated LCFAs, and in particular palmitic acid released as FFAs, are less soluble, have a melting point above body temperature and can form insoluble crystalline acid soaps [31]. Palmitic acid-calcium soaps were reported to cause gastrointestinal discomfort and associated symptoms in infants fed an infant formula containing vegetable oils rich in palmitic acid at the *sn*-1,3 positions such as palm oil [31]. These soaps result in lower absorption of palmitic acid and calcium and tend to harden stools and cause constipation [68]. Other symptoms, such as crying and sleep disruption, that could be a result of gastrointestinal discomfort caused by constipation, were reported to improve or be prevented by consumption of formula with increased levels of *sn*-2 palmitic acid. Other mechanisms may involve signal acylated molecules, such as palmitoyl and oleoyl ethanolamides, of the endocannabinoid system that contribute to regulation of sleep and pain sensitivity [68].

To reduce the risk of formation of palmitic acid-calcium soaps and constipation synthetic fat blends have been developed to more closely resemble the triglyceride structure of human milk, with palmitic acid predominantly at the *sn*-2 position [68]. With goat milk having nearly three times more palmitic acid in the *sn*-2 position than some vegetable oils [3], inclusion of goat milk fat to formula is an alternative and natural way to increase the level of *sn*-2 palmitic acid. In a small double-blind randomised controlled trial, stool frequency, measured as bowel motions per day, was greater in infants fed a whole goat milk formula compared with a whey-adjusted cow milk formula [21]. Another larger double-blind randomised controlled trial reported no difference in stool frequency in infants fed a whole goat milk formula with 60% milk fat and 40% vegetable oils or a whey-adjusted cow milk formula with vegetable oils only [23]. Stool frequency was lower in both formula groups than in the breastfed reference group, however neither formula was supplemented with prebiotics. In a prospective cohort study, infants fed goat milk formula had similar stool frequency, and to some extent similar stool consistency, to that of breastfed infants and higher stool frequency than infants fed cow milk formula between 0 and 4 months of age [69]. In a case series study, infants with symptoms of constipation when fed cow milk formula experienced less crying, and had less fat excreted in stools and a change in stool consistency after switching to a whole goat milk formula with 55% of goat milk fat [70]. Another small study looked at feeding goat milk formula or cow milk formula from 0-3 months to 6 months of age and reported no difference in fecal characteristics and bowel motion between the groups [71]. However, the goat formula used was a whey-dominant formula with vegetable oils only and no goat milk fat but included higher levels of *sn*-2 palmitic acid and prebiotics. In another randomised equivalence trial, infants fed a formula supplemented with 48% cow milk fat and MFGM had more watery stools than infants fed a formula with vegetable oils only but closer to the stool consistency of breastfed infants [72]. While limited, the evidence points to a positive impact of milk fat on gastrointestinal health.

#### 9.1.3. Cholesterol and Its Importance in Early Life

Cholesterol is a key component of cellular membranes and its role in infant nutrition has been overlooked. Cholesterol is a precursor for steroid hormones, oxysterols, vitamin D and bile acids [73,74]. Bile, synthesized in the liver, is a fluid composed of bile acids and cholesterol and plays a key role in fat digestion and absorption in the small intestine [74]. Dietary and endogenous cholesterol is absorbed in the upper part of the small intestine [73]. Some lactic acid bacteria such as *Lactobacillus* present in the small intestine can metabolize cholesterol into coprostanol, which has a poor intestinal absorption. Cholesterol metabolism by gut bacteria, mainly *Eubacterium* and *Bacteroides*, into coprostanol is most efficient in the colon and contributes to increased excretion of cholesterol and therefore reduced blood cholesterol levels [73,74]. Most of the bile acid pool is absorbed and recycled back to the liver, but 5% is not reabsorbed and passes to the large intestine to be biotransformed by gut bacteria. There is an increased need for de novo synthesis of bile acids from cholesterol due to bacterial metabolism of bile acid contributing to cholesterol excretion. Entrapment of cholesterol by gut bacteria is another pathway, whereby gut bacteria utilise cholesterol for growth or incorporate it into their cell membrane [73,74]. The combined pathways, i.e., cholesterol absorption and recycling, secretion in bile, microbial metabolism of cholesterol and bile acids and microbial entrapment of cholesterol, coordinate the regulation of blood cholesterol level and the cholesterol balance between absorption, excretion and synthesis in the body.

Milk cholesterol is mainly present in the free state and in the MFGM bilayer complexed with SM in tightly packed domains. Surprisingly SM, and particularly milk SM, has been shown to reduce cholesterol absorption [64,75]. The microstructure of the MFGM and the digestion of SM in the lower small intestine and large intestine [75] may be a way to limit cholesterol absorption in the small intestine and deliver cholesterol to the large intestine for microbial metabolism. The unique composition and structure of human milk, rich in cholesterol complexed with sphingomyelin, may have a programming effect to regulate cholesterol homeostasis throughout life.

With the shift to use only vegetable oils as the source of lipids in infant formulas, most formulas contain far less cholesterol than human milk [38] but also lower SM levels [49]. Infants fed formulas with low cholesterol levels have lower serum levels of total cholesterol, low-density lipoprotein cholesterol and higher rate of cholesterol synthesis than breast-fed infants [76,77]. Therefore, the use of whole goat milk as a source of milk fat and MFGM, including SM, may help regulate the cholesterol balance as human milk does. Indeed, simply adding cholesterol to infant formula did not significantly change cholesterol metabolism in early life [78]. In contrast, the supplementation of formula with an MFGM-rich whey ingredient led to an increase in total serum cholesterol level comparable to levels in the breastfed group but did not affect low-density lipoprotein to high-density lipoprotein ratio [79].

Another aspect in favour of increasing the levels of milk fat, and thereby reducing the levels of vegetable oils, in infant formula is the effect of vegetable oil-derived plant sterols on cholesterol absorption [38]. Plant sterols are known to reduce cholesterol absorption in adults. Plant sterols are present in trace amounts in human milk, likely derived from the maternal diet. Infant formulas manufactured with vegetable oils as the sole source of lipids contain much higher concentrations of plant sterols [38]. The role of plant sterols in infant nutrition is not yet known. While it is not possible to manufacture infant formula with 100% milk fat, blending goat milk fat and vegetable oils offers a way to maximise the level of cholesterol while minimizing the level of plant sterols.

#### 9.1.4. Size and Microstructure of the Milk Fat Globules

The complex structure of human milk fat globules likely plays a role in the digestion, absorption and targeted delivery of lipids within the infant’s GIT [36,68]. Most infant formulas have a different lipid structure as vegetable oils are homogenized with milk proteins resulting in milk protein-stabilised droplets with a smaller average diameter and more uniform size distribution than human milk fat globules [5]. The digestion of lipids is an interfacial process, whereby access to triglycerides is controlled by the size and the interfacial composition and structure of the lipid droplets [36]. In preterm infants, gastric lipolysis of human milk lipids was greater than that of infant formula lipids, likely due to the structure and composition of human milk fat globules facilitating access to the gastric lipase [66]. The presence of milk phospholipids at the interface of emulsions was shown to modulate both lipid and protein hydrolysis under in vitro gastric digestion and in piglets [36,80].

Improving the structure of infant formula lipid droplets by increasing their average diameter, stabilizing their interface with MFGM components and retaining some MFGM fragments within the aqueous phase [5] has been shown to modify the metabolic phenotype in infancy. Consumption of such formula in infancy resulted in an attenuation of excessive body fat accumulation induced by a western-style diet in adulthood in animal models, a finding yet to be confirmed in humans [72].

Many clinical studies investigating the role of the MFGM in early life have been published in the last decade. The studies have all been conducted using MFGM-enriched fractions from cow milk [65] and therefore may have different microstructures and phospholipid ratios as found in human milk [39]. However, one remaining question is whether the MFGM components play the same role when present at the interface of fat globules, as intact MFGM fragments in the aqueous phase or pulled apart after processing of MFGM-rich ingredients or infant formulas [39]. The accumulating evidence of an effect on outcomes such as metabolic, immune and cognitive development suggests that MFGM components retain their bioactivity during manufacturing and gastrointestinal processing [39,65].

### 9.2. The Role of Milk Lipids in Cognition and Brain Development

Breastfeeding, including mother-infant bonding and the nutritional value of human milk, has been shown to improve brain and cognitive development. While a lot of research has focused on the DHA content in human milk and its role in the development of the growing brain and cognitive functions, recent studies demonstrated that other milk lipids are involved [39,49,65]. The brain is a lipid-rich organ and the infant brain consumes up to 74% of the daily dietary energy intake [81], suggesting that the quality and quantity of dietary lipids are critical to optimal brain development in early life.

#### 9.2.1. The Role of Milk Lipids

MCFAs are oxidised rapidly and may prevent oxidation of (long chain)-polyunsaturated fatty acids ((LC)-PUFAs) and increase the conversion of PUFAs to LC-PUFAs, promoting increased LC-PUFA accumulation in the brain [2]. In a clinical study where infants were either breastfed or fed a formula with vegetable oils only or with a mixture of vegetable oils and dairy lipids, Gianni et al. [82] showed that the addition of dairy lipids led to an increased endogenous conversion of omega-3 LC-PUFAs from PUFAs, with higher total omega-3, DHA and DPA levels in red blood cells. They speculated that the higher levels of dairy lipid SCFAs and MCFAs, being rapidly absorbed and oxidised, may spare oxidation of PUFA precursors, favouring endogenous conversion to LC-PUFAs. They also hypothesized that it may support accretion of DHA, neuroplasticity and neurogenesis in the brain, as observed in animal models [82].

Infant plasma samples from the study by Zhou et al. [23] were analysed for glycerophospholipid species [83]. The breastfed group had higher levels of PC(16:0/16:0) and PC(18:0/16:0), reflective of the higher levels of *sn*-2 palmitic acid in human milk, and higher glycerophospholipids containing LC-PUFAs than goat and cow milk formula groups. The whole goat milk formula group had higher levels of glycerophospholipids containing myristic and palmitoleic acid and slightly higher PC(18:0/16:0) and glycerophospholipids containing LC-PUFAs than the cow milk formula group. It may be interesting to note that palmitoleic acid levels were found in greater concentration in human milk of mothers whose infant had cold-like symptoms [84]. Both formulas used in Zhou et al. [23] were not supplemented with LC-PUFAs, demonstrating that endogenous synthesis of LC-PUFAs from PUFAs alone was not sufficient to attain levels in breastfed infants. Of note, in this study plasma samples were analysed and are known to be more influenced by recent dietary intake than red blood cell membranes. Interestingly, while there were some differences between the breastfed, cow milk formula and goat milk formula groups, serum albumin, urea, creatinine, hemoglobin, ferritin and folate levels were within the normal range for 4 month-old infants and plasma amino acid levels in the formula-fed groups were comparable to that of the breastfed group [23].

A high ketone body turnover is observed within less than 8–10 h of fasting in infants [31]. Saturated MCFAs absorbed in the small intestine are rapidly transported to the liver where they are β-oxidized to acetyl-coA, an intermediate substrate in the biosynthesis of other fatty acids and ketone bodies (Novak and Innis, 2011). MCFAs are potential substrates for ketone bodies contributing to ketogenesis to support the growing brain between feeds [31,85]. Oxidation of C16:0 and C16:1*n*-7 (present in goat milk) can also potentially be converted to ketones to contribute to the development of the growing brain [31].

In the study by Zhou et al. [23], 4-month plasma samples with higher SM levels were observed in the breastfed group. However, SM(d18:1/17:0) concentrations were higher in the goat formula group, reflective of the presence of C17:0 in goat milk fat [86]. Carnitine ester levels were similar in the three groups, but the breastfed group had lower levels of butyl-carnitine, due to lower protein intake. C18-1-carnitine levels were similar in the goat formula and breastfed groups. Of note, the plasma SM species included mainly endogenously-synthesized SCFAs and MCFAs, which may be modulated by substrate preference of ceramide synthases [86].

The study by Timby et al. [37], where infants were either breastfed or fed a formula supplemented with an MFGM-enriched whey ingredient or a standard formula from 2 to 6 months of age, demonstrated that, while both formula-fed infant groups had a metabolic phenotype skewed towards protein metabolism, the metabolic phenotype of MFGM-enriched formula-fed infants had greater concentrations of fatty acid oxidation products [81]. Breastfed infants had even greater levels of fatty acid oxidation products, confirming a preference for fat metabolism. The higher levels of LCFAs, MCFAs, acylcarnitine species and ketone bodies in the serum of breastfed infants are indicative of elevated ketogenesis to support the rapidly growing brain. The differences between groups disappeared once complementary foods were introduced, suggesting an early metabolic programming effect partially directed by the MFGM [81].

The above results suggest that adding goat milk fat with high levels of SCFAs, MCFAs and *sn*-2 palmitic acid and MFGM to infant formulas could direct the metabolic phenotype of infant towards a preference for fat metabolism and ketogenesis, with potential beneficial effects for brain development.

#### 9.2.2. The Role of MFGM Lipids

The preclinical and clinical evidence for a role of MFGM polar lipids, with or without the addition of milk fat, in the development of the brain and cognitive function in infancy has dramatically increased in the last decade [37,39,49,87]. The hypothesis stems from the fact that MFGM and neuronal membranes share a similar composition, rich in phospholipids, sphingolipids and cholesterol. Thus, MFGM lipids have the potential to provide building blocks for the rapidly growing brain in the first years of life [39]. About 25% of cholesterol in the body is found in the brain, mainly in the form of myelin [88]. Cholesterol synthesis is an active process in the first weeks postpartum, contributing to neurite outgrowth, synaptogenesis, myelination and other key neurocellular processes. Sphingolipids are also key for optimal synaptogenesis, neurogenesis and myelination in early life [88,89]. Cholesterol and SM together promote brain maturation through myelination. Gangliosides represent 10% of lipids in the brain, principally in the cerebral cortex [90]. In a clinical study [91] where infants were fed an MFGM ganglioside-enriched formula, a standard formula or breast milk in the first 6 months of life. Supplementation with MFGM gangliosides resulted in cognitive development scores and serum ganglioside levels similar to those of the breastfed group and greater than those of the control group.

Supplying MFGM from 2 to 6 months of age raised serum myo-inositol and choline and downstream products and LysoPCs [81]. Serum choline was also increased after supplementation with MFGM in 6-11 month-old infants [92]. SM and PC are a source of choline, an important nutrient in brain cell membranes and for the synthesis of neurotransmitters such as acetylcholine, neurogenesis and synaptogenesis [40]. MFGM supplementation between 2 and 6 months of age resulted in higher concentration of SM, PC and ceramide species in plasma at 6 months of age, higher concentration of SM and PC species in serum at 4 months of age but not at 12 months of age, and higher concentrations of SM, PE and PC species in erythrocyte membranes at 6 months of age but not at 12 months of age [93]. These results support a role for MFGM lipids in immune and cognitive outcomes reported by [37,58].

While having different study designs and MFGM-rich ingredients with or without other formulation differences (e.g protein levels or addition of specialty ingredients), several clinical studies in infants have demonstrated an improvement in cognitive functions after supplementation of formula with MFGM fractions in the first 6-12 months of life [37,39,49,65,87]. It is however not possible to associate the improvement in cognition to a single MFGM component and it is more likely that the effect is due to a combination of MFGM components, in particular PC, PS, SM, gangliosides and cholesterol.

### 9.3. Development of the Immune System

The mucosal immune system of infants is immature at birth. Human milk, being the sole source of nutrition in the first 4–6 months of life, plays a critical role in training the immune system and the maturation of the gut, such as the development of the gut microbiome. The composition and structure of the human milk fat globules not only provide a quick release of energy but also a controlled hydrolysis of the milk fat globules for a delivery of bioactive components along specific sites of the GIT to contribute to immune education and primo-colonization [36]. Milk lipids and in particular MFGM lipids further contribute to the improvement of the intestinal epithelial barrier functions and structural integrity by providing building blocks for mucosal cell membranes [40].

#### 9.3.1. Milk Lipids Have Antimicrobial Properties

In vitro and in vivo studies have highlighted antimicrobial properties of SCFAs and MCFAs against *Escherichia coli*, *Listeria monocytogenes* and *Staphylococcus aureus* [3]. Supplementation of formula with gangliosides, present in the MFGM, decreased levels of fecal *Escherichia coli* and increased levels of *Bifidobacteria* in preterm newborns [94]. As demonstrated for human milk, lipolytic products from milk fat digestion have detergent-like properties and a protective role against viruses, bacteria, fungi and protozoa through disruption of their lipid membranes [67].

#### 9.3.2. The Role of MFGM

Despite cow and goat MFGM proteins having a less diverse profile than human MFGM, they include the same major proteins. While MFGM proteins represent only up to 4% of total milk proteins, they have biologically relevant functions, including membrane and protein trafficking, cell signaling and immune functions [58,65]. Glycans on glycosylated proteins act as decoy receptors for pathogens and viruses within the GIT, limiting their adherence to the GIT wall but also preventing their proliferation [64,67]. Xanthine oxidase, butyrophilin, MUC1, lactadherin, CD14, and toll-like receptor 1 and 4 have antimicrobial properties [58]. MFGM proteins have been shown to regulate gut inflammation. For example, osteopontin was demonstrated to balance the Th1 and Th2 immune responses by controlling cytokine production [40]. In addition, thanks to their high degree of glycosylation, the glycoproteins MUC 1 and lactadherin are resistant to digestion in the infant’s stomach and to some extent in the infant’s intestine, meaning they can reach the small intestine to exert their protective function [67]. Lactadherin, abundant in the goat MFGM, has potential effects in the maintenance of intestinal epithelial homeostasis and mucosal repair [63] and as potent antiviral agent [62].

MFGM sphingolipids such as sphingomyelin, gangliosides and glycosphingolipids, either directly or through their metabolites, have immunomodulatory effects and a protective effect against pathogens [89]. Sphingolipids are not digested in the upper gastrointestinal tract [36]. For example, sphingomyelin is hydrolysed into ceramide and sphingosine by sphingomyelinase in the lower half of the small intestine, resulting in sphingomyelin and its metabolites being active in the small intestine and the colon, where they exert bactericidal and anti-inflammatory activity [36,89]. Protection against pathogens is mainly driven by the presence of glycans on MFGM glycolipids, decreasing pathogens adherence to the intestinal mucosa [89].

Le Huërou-Luron et al. [80] demonstrated that maturation of the immune system in piglets fed an infant formula with milk fat and MFGM was more similar to that of piglets fed sow milk than a formula with vegetable oils only. The enrichment in milk fat and MFGM also led to an increase in fecal *Proteobacteria* and *Bacteroidetes*, a decrease in *Firmicutes* and a higher abundance of *Parabacteroides*, *Escherichia/Shigella* and *Klebsiella* compared with the standard formula. Supplementation with MFGM in infancy and childhood was shown to reduce the incidence of acute otitis media, the number of episodes of bloody diarrhea, the number of days with fever and the use of antipyretics and to lower the concentration of serum immunoglobulin G against pneumococcus vaccine [58]. Lower incidences of respiratory adverse events and diarrhea were reported in Chinese infants fed a formula supplemented with MFGM and lactoferrin for 12 months compared to a non-supplemented formula [87]. 6–11 month-old Peruvian children whose diet was supplemented with MFGM for 6 months had lower prevalence of diarrhea and reduction in episodes of bloody diarrhea, and a decreased Th1 immune response due to reduced serum interleukin-2 levels than the non-supplemented group [92,95]. The preclinical and clinical effects indicate that formulas with both milk fat and MFGM can support a healthy immune system in infancy when breastfeeding is not possible.

#### 9.3.3. Modulation of the Gut Microbiota

A few hypotheses, namely the hygiene hypothesis, the old friends hypothesis and the biodiversity hypothesis, have been brought forward to explain the rise in allergies and have in common the impact of the environment, diet and lifestyle on the microbiota [96]. Several studies point to the critical contribution of early life microbial exposure and colonization in the development of food allergy, cutaneous and respiratory allergic diseases [97].

As described previously, some milk lipids, such as cholesterol, polar lipids and fatty acids, can modulate the composition of the gut microbiota. A 6-week consumption of an infant formula with increased level of *sn*-2 palmitic acid resulted in increased fecal abundance of *Lactobacilli* and *Bifidobacteria* [40]. Fermentation in the gut by *Bifidobacteria* leads to the release of metabolites, such as SCFAs, with immune-modulating and allergy-protective effects [96]. The fatty acid chain length and degree of saturation influence the gut microbiota. C10:0 and C12:0, present in human milk lipids, can inhibit the growth of food-borne pathogens, and the release of lipolytic products with detergent-life properties during digestion of human milk contribute to the antimicrobial effect of milk fat [40].

Glycosylated lipids and proteins from the MFGM are able to, at least partially, resist digestion in the upper GIT and therefore can be utilised as prebiotics by commensal gut bacteria in the distal gut [40]. However, in the study by Timby et al. [37], there were only minor differences in the fecal microbiome of infants fed either the lower-protein, MFGM-supplemented formula or the standard formula, and both groups had lower abundance of *Bifidobacteria* than the breastfed group before the introduction of solid foods [81]. In addition, the breastfed group had a more heterogenous fecal metabolome and the lower-protein, MFGM-supplemented formula group had lower levels of fecal amino acid and their derivatives, lactate and succinate than the standard formula group [81]. This indicates that MFGM supplementation alone is not sufficient to shift the fecal microbiome and metabolome closer to that of breastfed infants and that a combination of milk fat and MFGM may be more advantageous.

The stool microbiota of infants fed whole goat milk formula from birth was more similar to that of breastfed infants than was that of infants fed cow milk formula at 2 months of age [98]. As expected, the stool samples of breastfed infants had greater abundance of *Bifidobacteriaceae* than stools from formula-fed infants. Vaginal delivery also correlated with greater *Bifidobacteriaceae* abundance. *B. longum*, *B. breve* and *B. bifidum* were the most abundant bifidobacterial in all three groups. Goat formula-fed infants and breastfed infants had a simpler composition of *Lachnospiraceae* than cow formula-fed infants [98].

#### 9.3.4. Can Milk Lipids Mediate the Development of Allergies?

To date, there have been only animal studies or small clinical studies in pediatric populations providing some evidence for a role of goat milk in the reduction of allergy risk in infancy [99,100,101]. Some children previously sensitised to cow milk showed less severe reactions to goat milk [99,102]. The development of allergy involves several mechanisms, and, while the type of proteins may underlie the differential response to goat and cow milk, the role of proteins will not be covered in this review. Instead, we focus on the evidence that lipids may play a role in the prevention or reduction of the risk of allergy in early life as proposed for human milk lipids [90].

There is emerging evidence of a role of fatty acids in the development of allergy in early life. For example, SCFAs, either dietary, from carbohydrate fermentation by gut bacteria or delivered via the placenta during pregnancy, have been shown in animal models and in humans to lower the risk of developing allergic and atopic diseases [28]. After absorption in the GIT, SCFAs contribute to various functions such as regulatory T cell responses and tolerance, synthesis of dendritic cell precursors in the bone marrow, and secretion of mucus and promotion of epithelial barrier integrity in the gut. Interestingly, it was reported that atopic mothers had lower concentration of the SCFAs, acetate and butyrate, in mature milk than non-atopic mothers [28]. A lower exposure of dietary SCFAs in infancy could contribute to atopy and overweight risk in later life.

Trans fatty acids (TFAs) are a controversial type of fatty acids but are naturally occurring in human milk and ruminant milk, such as goat milk. Vaccenic acid and CLA are the two most studied milk TFAs. Milk TFAs are different from TFAs formed during hydrogenation of vegetable oils [103]. Higher TFA levels in human milk or circulating levels during pregnancy have been correlated with lower risk of atopic diseases. CLAs have also been shown to have immunomodulatory properties, such as production of cytokines and nitric oxide, which may influence the development of atopy [103]. Vaccenic acid and CLA from ruminant milk can potentially reduce allergic inflammation and sensitization, via reduction of eicosanoid precursors and PPAR-gamma-related mechanism [104]. Current international regulations on the composition of formulas for infants and young children only allow a maximum TFA level of 3%, without distinction between milk TFAs and industrial TFAs.

MCFAs, by preventing oxidation of PUFAs and enhancing their bioconversion into LC-PUFAs, could also indirectly contribute to the incorporation of essential omega-3 and omega-6 fatty acids and their metabolites in the lipid-rich extracellular matrix of the stratum corneum [104]. Dietary supplementation with PUFAs, and especially with DHA, in animal models showed beneficial effects on allergic symptoms in the gut, skin and lung. There is also increasing evidence of a preventive effect of supplementation with *n-3* (LC)-PUFAs with or without *n-6* (LC)-PUFAs against allergic diseases in infants [104]. Thus, inclusion of MCFAs in infant formula, with or without LC-PUFAs, may have an indirect preventive effect against allergy.

A continuous lipid structure, rich in ceramide, cholesterol and FFA, in the stratum corneum act as skin barrier to xenobiotics [89]. Alteration to this lipid structure results in compromised epidermal functions and skin diseases such as atopic dermatitis. A strong skin barrier is required to prevent the penetration of microbes, allergens and irritants, and the subsequent activation of type-2 immune and inflammatory responses [105]. Clinical observations have concluded that skin lipid metabolism is critical in atopic dermatitis independent of the filaggrin genotype [105]. Skin lipids are critical for skin barrier integrity but also as antimicrobial and anti-inflammatory agents. Skin lipids in atopic dermatitis subjects tend to have disorganized lamellar membrane structures, different classes of ceramides as well as reduced fatty acid chain length in ceramides, and polar lipids [105]. Such disruption in the skin lipid structure leads to transepidermal water loss and xerosis in atopic dermatitis skin. Several studies have reported the role of dietary sphingolipids, ceramides and phospholipids, including from milk, to promote skin barrier functions and integrity, such as hydration, elasticity, lipid composition, reduced inflammatory status and transepidermal water loss [89,106,107,108].

In a recent lipidomics study [109], goat milk was found to contain higher levels of phospholipids than human and bovine milk, in particular greater levels of PC, SM, ceramide and hexosylceramide. Of note, human milk had more SM with ultra-long chain-fatty acids (ULCFAs) than cow milk but less than goat milk. Ceramides with ULCFAs are prevalent in healthy skin [105]. Milk glucosylceramides, galactosylceramides and lactosylceramides are hydrolysed to ceramides in the gut. Thus SM, ceramides, glycosphingolipids and gangliosides can be utilised for the maturation of the mucosal brush border membrane and brain cells but also as a potential source of ceramides in the skin [75]. Optimal absorption of palmitic acid from human milk provides an excess of palmitic acid for de novo synthesis of dihydrosphingosine, which is further converted to ceramides [75]. N-myristoylation and palmitoylation depend on the bioavailability of myristic and palmitic acids respectively and are critical for various T cell functions [104].

## 10. Conclusions

The composition and structure of the human milk fat globules for infant growth and development are now better understood and are drivers for the improvement of breast milk substitutes. Using whole goat milk in the manufacture of infant formulas supplies both milk fat and MFGM components thereby providing a lipid composition and structure closer to that of human milk fat and the potential to match more closely the physiological outcomes provided by human milk lipids. Clinical research in pediatric nutrition on goat milk-based formulas is very limited and mainly on infant formulas manufactured from fresh whole goat milk with a 20:80 whey:casein ratio. Further research is warranted to confirm or understand the potential of whole goat milk for infant nutrition. In addition, one must note the difference in goat milk-based formulas available on the market, whether there are made from fresh whole goat milk without adjustment of the whey:casein ratio (i.e., providing milk fat and MFGM) or from goat milk-derived powdered ingredients often with an adjustment of the whey:casein ratio (i.e., with low to no milk fat and MFGM). The increasing number of studies demonstrate the advantage of supplying both milk fat and/or MFGM in early life to promote the healthy development of the metabolic response, the gut and its microbiota, but also for the optimal development of the brain and cognitive functions. The inclusion of whole goat milk as a source of both milk fat and MFGM in infant formula has the potential to contribute to the optimal gut–brain axis in early life. There is promising evidence that it may also play a part in modulating the gut–skin axis. Large clinical studies are required to confidently ascertain the preventive effect of a formula against the development of allergic diseases.

## Figures and Tables

**Figure 1 nutrients-12-03486-f001:**
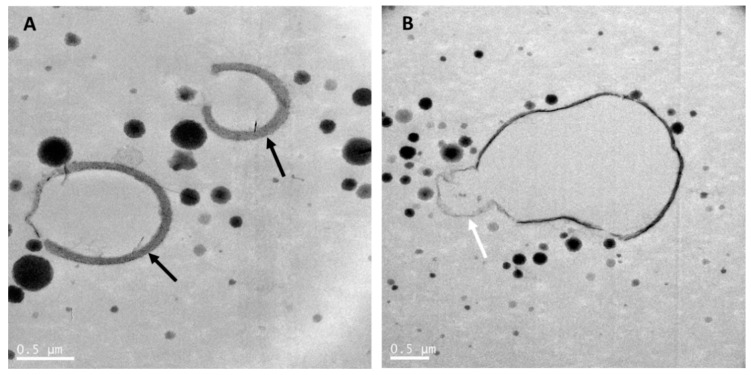
Transmission electron microscopy images of raw goat milk. (**A**) showing goat milk caseins and fat globules with a thick MFGM. (**B**) showing a milk fat globule with a cytoplasmic crescent (white arrow). Black arrows point at the MFGM surrounding fat globules. Goat milk was collected from a single goat in a New Zealand goat farm and kept at room temperature until processing within 2 h using the same method as in Gallier et al. [5]. Scale bar = 0.5 µm.

**Figure 2 nutrients-12-03486-f002:**
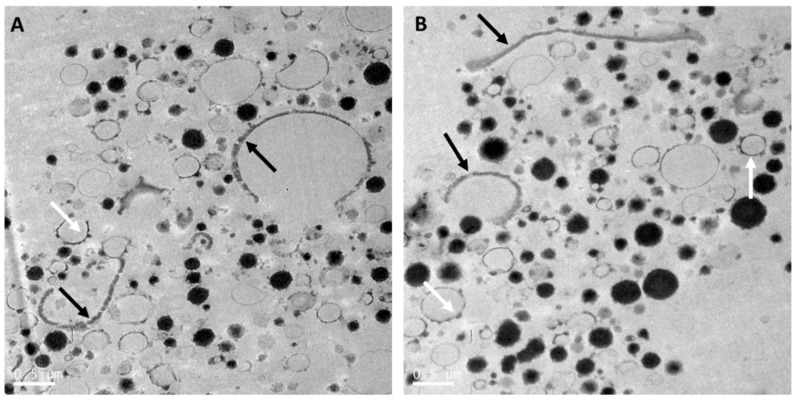
Transmission electron microscopy images (**A** and **B**) of reconstituted whole goat milk powder (Dairy Goat Co-operative, Hamilton, New Zealand) processed using the same method as in Gallier et al. [5]. Black arrows point at the MFGM surrounding fat globules (**A**) or as fragments in the serum phase (**A** and **B**). White arrows point at fat globules with adsorbed small casein micelles (**A** and **B**). Scale bar = 0.5 µm.

**Figure 3 nutrients-12-03486-f003:**
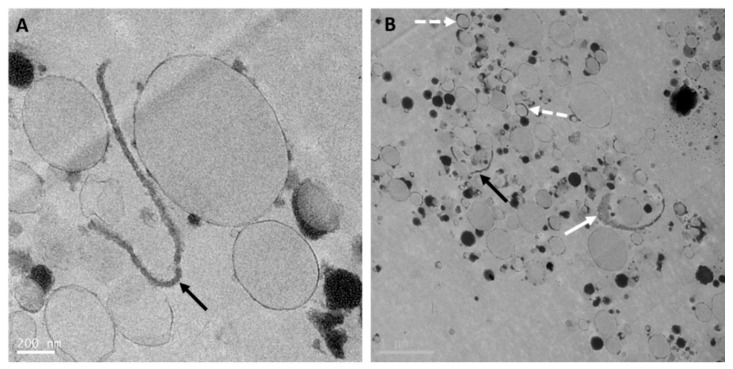
Transmission electron microscopy images (**A** and **B**) of reconstituted whole goat milk-based infant formula (Dairy Goat Co-operative, Hamilton, New Zealand) processed using the same method as in Gallier et al. [5]. Black arrows point at MFGM fragments in the serum phase (**A** and **B**). Full white arrow points at an MFGM fragment with attached cytoplasmic crescent filled with electron-dense material (**B**). Dashed white arrows point at droplets with thicker interface (**B**). Scale bars = 200 nm (**A**) and 1 µm (**B**).

**Table 1 nutrients-12-03486-t001:** Fatty acid concentration (% of total fatty acids) in human milk, cow milk, goat milk and whole goat milk-based infant formula.

	Human Milk ^1^ Europe	Human Milk ^1^ Asia	Cow Milk ^2^	Goat Milk ^2^	Whole Goat Milk-Based IF ^3^ 48% MF	Whole Goat Milk-Based IF ^4^ 55% MF	Cow Milk-Based IF Vegetable Oil Only ^6^	Cow Milk-Based IF MF ^1^
Butyric acid C4:0	ND	ND	3.2–3.3	2.0–2.6	1.17	3.1	ND	2.4
Caproic acid C6:0	0.39	0.07	1.6–2.1	2.4–2.9	1.06	2.5	ND/0.2	1.3
Caprylic acid C8:0	0.19 (0.09–0.24)	0.17 (0.11–0.28)	1.2–1.3	2.7–2.7	1.11	2.0	1.2/2.5	1.7
Capric acid C10:0	1.29 (0.83–1.63)	1.31 (0.52–2.48)	3.0–3.1	8.4–9.7	3.43	7.3	1.1/1.8	2.2
Lauric acid C12:0	5.98 (4.15–8.33)	5.56 (2.97–13.82)	3.1–3.3	3.3–4.3	1.54	4.2	5.4/13.4	6.3
Myristic acid C14:0	6.44 (4.98–9.38)	5.70 (3.50–12.12)	9.5–12.1	9.6–10.3	3.68	7.0	4.6/5.2	7.2
Myristoleic acid C14:1	0.18	0.26 (0.03–1.11)	0.7–1.1	0.09–0.16	0.12	ND	ND/ND	0.8
Pentadecanoic acid C15:0	0.25 (0.16–0.32)	0.20 (0.08–0.50)	ND	ND	0.35	0.6	ND/ND	0.6
Palmitic acid C16:0	21.93 (15.43–25.62)	21.78 (17.55–29.00)	26.5–32.2	24.6–27.7	12.30	17.0	26.3/7.7	18.9
Palmitoleic acid C16:1 *n-7*	1.98 (1.65–2.31)	2.44 (1.29–4.59)	ND	ND	0.39	ND	0.6/0.1	1.1
Heptadecanoic acid C17:0	0.29 (0.22–0.33)	0.28 (0.19–0.41)	ND	ND	0.29	0.4	ND/ND	0.3
Stearic acid C18:0	7.37 (5.58–9.52)	5.58 (3.90–6.79)	8.9–14.6	9.7–12.5	5.89	6.3	5.3/3.2	6.7
Oleic acid C18:1 *n-9*	36.30 (28.93–41.69)	30.80 (21.85–36.96)	19.3–24.1	19.4–24.0	40.65	31.0	37.6/43.3	28.1
Linoleic acid C18:2 *n-6*	13.99 (10.16–16.59)	16.90 (7.53–24.29)	ND	ND	10.79	14.0	14.0/20.5	16.7
Conjugated linoleic acid C18:2 *c*9, *t*11	0.27–0.49 ^5^	ND	0.1–1.9	0.4–3.7	0.33	ND	ND/ND	ND
α-linolenic acid C18:3 *n-3*	0.76 (0.49–1.05)	1.47 (0.35–4.06)	ND	ND	1.58	1.2	1.6/1.8	1.5
Arachidic acid C20:0	0.21 (0.14–0.31)	0.32 (0.03–2.97)	ND	ND	0.24	0.3	ND/0.3	0.3
Arachidonic acid C20:4 *n-6*	0.47 (0.37–0.64)	0.64 (0.30–2.57)	ND	ND	0.45	ND	0.3/0.3	ND
Eicosapentaenoic acid C20:5 *n-3*	0.09 (0.05–0.13)	0.31 (0.07–1.59)	ND	ND	0.12	ND	ND/0.0	ND
Behenic acid C22:0	0.09 (0.05–0.13)	0.08 (0.05–0.14)	ND	ND	0.33	ND	ND/0.4	0.1
Docosahexaenoic acid C22:6 *n-3*	0.28 (0.18–0.42)	0.55 (0.19–1.13)	ND	ND	0.44	ND	0.2/0.2	ND
Tetracosanoic acid C24:0	0.07 (0.03–0.16)	0.07 (0.01–0.14)	ND	ND	0.21	ND	ND/0.1	ND

^1^ from [3]. ^2^ from [26]. ^3^ Measured using gas chromatography (*n* = 2). ^4^ from [25]. ^5^ from [29]. ^6^ from [3]; values (%/%) are for cow milk-based IF manufactured with a blend of vegetable oils with palm oil/without palm oil. ND: not determined. IF: infant formula. MF: milk fat.

**Table 2 nutrients-12-03486-t002:** *sn*-2 palmitic acid percentage in human milk, cow milk, goat milk and whole goat milk-based infant formula.

	Human Milk ^1,2^	Cow Milk ^2,3^	Goat Milk ^3^	Whole Goat Milk-Based IF ^4^ 48% MF	Whole Goat Milk-Based IF ^3^ 55% MF	Cow Milk-Based IF Vegetable Oils Only ^2^
% *sn*-2 palmitic acid	51–88	37–45	35	29.5	31	10–20/39–47 ^5^

^1^ from [6,35]. ^2^ from [3]. ^3^ from [25]. ^4^ Measured with ISO 6800 by ITERG, France (*n* = 6). ^5^ Values (%/%) are given for cow milk-based IF manufactured with a blend of vegetable oils without structured palm oil/with structured palm oil. IF: infant formula; MF: milk fat.

**Table 3 nutrients-12-03486-t003:** Cholesterol concentration (mg/L) in human milk, cow milk, goat milk and whole goat milk-based infant formula.

	Human Milk ^1^	Cow Milk ^2^	Goat Milk ^3^	Whole Moat Milk-Based IF ^4^	Cow Milk-Based IF ^5^
Cholesterol (mg/L)	90–200	100–300	100–200	58	14.6–51

^1^ from [40,41]. ^2^ from [3,41]. ^3^ from [30]. ^4^ Measured using gas chromatography with flame ionization detector (AOAC 970.51) (*n* = 6). ^5^ from [38] from a range of commercial IFs with or without the addition of cow milk fat and MFGM. IF: infant formula.

**Table 4 nutrients-12-03486-t004:** Total phospholipid concentration (mg/L) and percentage of phospholipid species (% of total phospholipids) in human milk, cow milk, goat milk and whole goat milk-based infant formula.

	Human Milk ^1^	Cow Milk ^3^	Goat Milk ^3^	Whole Moat Milk-Based IF ^4^
Total PL (mg/L)	286.6 (98–474 ^2^)	294–400 ^5^	276 ^5^	169.2
PL (mg/L)				
PI	2.2–21	0.5–26.7 ^6^	21.1 ^4^	17.5
PC	32–124	18.2–128.0 ^6^	93.7 ^4^	51.6
PS	11–45	2.5–56.5 ^6^	38.7 ^4^	18.3
PE	26–103	19.2–143.0 ^6^	91.6 ^4^	40.2
SM	25–177	11.9–98.9 ^6^	91.2 ^4,7^	38.7 ^7^
% of total phospholipids				
PI	1.1–10	0.1–9.0	2.21–9.4	11
PC	19–38	25.9–33.2	27.4–31.6	31
PS	3.7–17	0.12–9.1	2.41–14.0	11
PE	5.9–36	23.4–46.7	26.9–46.1	24
SM	29–43	19.8–25.4	16.1–27.3	23 ^7^

^1^ from [38]. ^2^ from [49]. ^3^ from [26]. ^4^ Measured using ^31^Phosphorus-nuclear magnetic resonance (*n* = 3 for goat milk and *n* = 5 for IF). ^5^ from [51]. ^6^ from [52]. ^7^ sum of SM and DHSM (dihydrosphingomyelin). PL: phospholipids; PI: phosphatidylinositol; PC: phosphatidylcholine; PS: phosphatidylserine; PE: phosphatidylethanolamine; SM: sphingomyelin.

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
