# Peer review of "Whole Goat Milk as a Source of Fat and Milk Fat Globule Membrane in Infant Formula"

_nutrients, 2020, doi:10.3390/nu12113486_

Round 1

Reviewer 1 Report

Thank you for giving me an opportunity to review this manuscript.

In this review authors focused on the infant, follow-on and young child formulas based on whole goat milk without adjustment of the whey:casein ratio (i.e. maintaining the natural 20:80 whey:casein  ratio of whole goat milk) and with clinical evidence that it supports healthy growth and  development of the infant.

During infancy, appropriate nutrition is required for normal growth and development. There is now substantial evidence to indicate that early nutrition also has fundamental implications for long-term health by programming aspects of ensuant cognitive function, preventing obesity and anaemia, as well as reducing the risk of serious diseases.
Where breastfeeding is not possible or breast milk is insufficient, infant formulas are used. Traditionally, infant formulas made with cow’s milk are the first line choice for formula-fed infants, so until recently most of the internationally published conclusive studies in human infants have been limited to the evaluation of cow’s milk or soy protein-based infant formulae. The use of goat’s milk formula is a case in point.

This a interesting paper.

Comments for authors:

Abstract: the aim of the study need to be more clear

Line 69 please ad the reference: Leong A, Liu Z, Almshawit H, Zisu B, Pillidge C, Rochfort S, Gill H. Oligosaccharides in goats' milk-based infant formula and their prebiotic and anti-infection properties. Br J Nutr. 2019 Aug 28;122(4):441-449

Section: 6. Whole goat milk as a source of MFGM polar lipids

Please ad the reference: Manoni M, Di Lorenzo C, Ottoboni M, Tretola M, Pinotti L. Comparative Proteomics of Milk Fat Globule Membrane (MFGM) Proteome across Species and Lactation Stages and the Potentials of MFGM Fractions in Infant Formula Preparation. Foods. 2020 Sep 7;9(9):1251.

Author Response

We thank the reviewer for the feedback. We have added lines 22-24 to state the aim of the review.

The Leong et al. reference has not been added for the following reasons:

  • The study was on a whey-adjusted infant formula whereas the present review deals with non-whey-adjusted formulas.
  • The study did not demonstrate safety in infant populations and it was an in vitro study using oligosaccharide fractions from goat milk formula to stimulate the growth of specific bacteria. Oligosaccharides are outside of the scope of the present review.

The Manoni et al. reference has been added as reference 61. Please note it was added in the section on MFGM proteins, and not MFGM lipids.

Reviewer 2 Report

The submitted manuscript presents an extensive and careful review on a relevant topic and with high clinical applicability, namely in advising breast milk substitute formulas when breastfeeding is not possible.

The manuscript is well written and focuses on relevant and current aspects, such as the role of milk in the development of the intestinal microbiota and in immunomodulation.

However, it would be important for the authors to address two questions:

1- In the comparative tables it would be important to see the comparison with cow milk-based infant formula, taking into account that these alternatives are mostly offered to children and not cow's milk in nature.

2 - Although the manuscript focuses exclusively on the issue of fat, it cannot be inseparable from more general recommendations from the point of view of public health. So it would be very important to see a reference / review / discussion regarding the other nutrients present in goat's milk such as proteins, carbohydrate content, namely lactose, specific vitamins and minerals (such as iron and folic acid - and their bioavailability), taking into account that the recommendations for ingesting goat's milk in cow's milk, for example, take the entire nutritional profile into account.

Author Response

We thank the reviewer for the feedback.

  1. This is a good comment and something we had considered. However, cow milk-based infant formulas are very variable from one manufacturer to the other. For example, some will use only skim milk and whey protein powders; the whey protein type can also vary (i.e. demineralized whey, whey protein concentrate, sweet whey...); some will use whole milk powder instead of skim milk; some will add cream or anhydrous milk; some will use a blend of vegetable oils only as the source of lipids, with or without palm oil or with structured lipids (i.e. OPO)... All these factors will impact the lipid composition and therefore there is not one typical composition of cow's milk formulas. Hence we had decided to make reference to literature values within the text where relevant and appropriate for the discussion (please see lines 110-112, 126-129, 140-146, 198-201, and 219-222 in the revised manuscript).
  2. We appreciate that the review mainly focuses extensively on goat milk fat and that goat milk provides more nutrients. However, goat milk-based formulas, similarly to any other type of formulas, need to adhere to strict compositional regulations aimed at matching the composition of human milk as close as possible. Therefore the formulas are supplemented with lactose (as human milk has a higher content than cow and goat milk), vitamins and minerals. Dr Colin Prosser, one of the co-authors has drafted a separate manuscript on goat milk and its composition going beyond its role for infant nutrition.